# Fatigue Crack Growth from Notches: A Numerical Analysis

**Micael Borges** , **Manuel Caldas, Fernando Antunes \*, Ricardo Branco and Pedro Prates**

Department of Mechanical Engineering, Centre for Mechanical Engineering, Materials and Processes (CEMMPRE), University of Coimbra, 3030-788 Coimbra, Portugal; micaelfriasborges@outlook.pt (M.B.); manuel_luis6@hotmail.com (M.C.); ricardo.branco@dem.uc.pt (R.B.); pedro.prates@dem.uc.pt (P.P.)

\* Correspondence: fernando.ventura@dem.uc.pt; Tel.: +351-790-700

**Abstract:** A numerical approach based on plastic crack tip opening displacement (CTOD) was followed to study fatigue crack growth (FCG) from notches. The identification of fundamental mechanisms was made considering notched and unnotched models, with and without contact of crack flanks. Different parameters were studied, namely, notch radius, crack length, stress state, and material. The notch increases the plastic CTOD, and therefore fatigue crack growth rate, da/dN, as expected. The reduction of notch radius increases da/dN but reduces the notch affected zone. Ahead of the notch affected zone, da/dN increases linearly with crack growth, with a rate that increases linearly with the plastic CTOD. The crack closure phenomenon has a dramatic effect under plane stress conditions but a limited effect on plane strain conditions. In the former case, the contact of crack flanks reduces substantially the effect of notch radius and the size of the notch affected zone. These trends are associated with the increase of crack closure level with notch radius. The material does not affect the global trends, but the reduction of yield stress increases the level of plastic deformation and therefore da/dN. The effect of material, and also of stress state, is mainly associated with crack closure.

**Keywords:** fatigue crack growth; notch; notch radius; CTOD; crack closure

## 1. Introduction

In components submitted to cyclic loads, notches act as stress concentrators promoting the occurrence of fatigue failure. An understanding of the effects of notches is therefore fundamental for the proper design of structural components. At a blunt notch, crack nucleation represents the major part of fatigue life, and nominal stress approaches, local stress-strain approaches, critical distance theories, or weighting control parameters-based approaches are used for design [1]. On the other hand, at a sharp notch, the crack nucleates quickly due to the high stresses at the notch root. The propagation phase is dominant, and a fracture mechanics-based approach is usually considered for design. This approach can also be used to predict initiation life, assuming a crack at the notch tip and calculating the number of load cycles required to propagate the crack up to a size detectable by nondestructive inspection methods.

In fatigue crack growth (FCG) from notches, there is a confluence of different relevant topics, namely, stress and strain fields at notches, short cracks, and crack closure. There is a great complexity associated with the plasticity induced by the notch, the crack propagation across the stress-strain fields produced by the notch, and the progressive formation of residual plastic wake. Once the crack grows through the notch-influenced region, it may stop growing [2,3]. After the crack exceeds a certain length, the notch geometry has little influence on the FCG of the advancing crack.

Different approaches have been followed to study FCG from notches. The stress intensity factor, K, is an elastic parameter with a long dominance in the study of FCG, being also widely used in the case of the notches. Ranganathan et al. [4] predicted fatigue lives in AA2024-T351 by integration of da/dN-ΔK laws. Several researchers attempted to modify the K approach to account for the effect of the notch [5–7]. Dowling [8] proposed an effective K model that accounts for notch effects on small cracks by modifying the remotely applied stress term by an appropriate stress concentration factor. However, the FCG rate is usually higher than expected considering da/dN-ΔK curves obtained in standard specimens. This odd behaviour was observed in carbon steels [9], stainless steel [10], and aluminum alloys [11]. Tanaka and Nakai [2] proposed that crack closure was the major cause. Generally, this short crack growth effect increases with the decrease of R and the increase of notch severity [12]. Sadananda and Vasudevan [13] modeled the short crack behavior by considering the stress intensity factors introduced by both the applied load and the residual stresses caused by the notch. Dong et al. [14] introduced a notch-induced magnification factor to modify ΔK when the crack was short. Vena et al. [15] introduced an effective ΔK to study the short cracks by considering the toughening effect in the phase transformation materials.

The plasticity associated with the notch may invalidate the use of linear elastic fracture mechanics [16], and approaches based on nonlinear parameters were exploited. Sehitoglu [17] found that the range of the J-integral could account for crack growth in the vicinity of notches. Merah et al. [18] used a finite element analysis to determine the J-integral for the simulation of the crack growth behavior near notches to consider the notch-induced plasticity. Wang and Lu [19] and Savaidis et al. [20] replaced the range of the J-integral by an effective value with the consideration of crack closure. Hammouda et al. [21] and Smith and Miller [22] attributed the observed short crack growth phenomenon to the combined effect of the notch plasticity and the crack-tip plasticity. Li [23] suggested that FCG from a notch was dominated by notch plasticity within the notch plasticity zone and the notch plasticity induced crack closure out of the notch plasticity zone. Based on the cyclic plasticity deformation obtained from the FE analysis, Ding et al. [12] modeled the elastic–plastic behavior of 1070 steel and used a damage accumulation criterion to predict da/dN. When the R-ratio was positive, the crack growth was mainly influenced by the plasticity created by the notch. When the R-ratio was negative, the contact of the cracked surfaces during a part of a loading cycle reduced the cyclic plasticity of the material near the crack tip. In previous works of the authors, FCG in standard specimens was studied using the plastic crack tip opening displacement (CTOD). This approach assumes that crack tip plastic deformation is the driving force and that the plastic CTOD is able to quantify this deformation. Crack closure is included in a natural way, and small-scale and large-scale yielding are studied without distinction. This approach was used to study FCG in 6082-T6 [24], 7050-T6 [25], and 2050-T8 [26] aluminum alloys, in the 18Ni300 steel [27] and in titanium [28], to predict the effect of stress ratio, stress state, and variable amplitude loading [29]; to predict fatigue threshold [30], and to predict the effect of material parameters [31].

The main objective here is to better understand the FCG from notches with the identification of the subjacent mechanisms. A numerical approach was followed to predict the FCG rates based on the plastic CTOD, assuming that this parameter is the crack driving force. In order to understand the notch effect, a comparison of cracks with and without notches is made. For the purpose of understanding the relevance of crack closure, models with and without contact of crack flanks were tested. These studies were made for different materials, notch radii and stress states, in order to understand the effect of these parameters in this context. Figure 1 is a schematic view of the paper, with the indication of the sections where the comparisons are made.

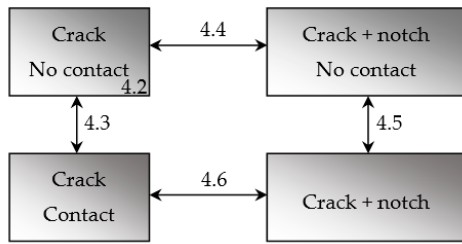

**Figure 1.** Schematic view of the study.

## 2. Material Model

The materials studied were the 6082-T6 and 7050-T6 aluminum alloys. These materials were selected because they are available in the laboratory for future experimental work and because their elastic–plastic behavior is characterized. A mixed elastic–plastic model was assumed, being the isotropic elastic behavior given by the generalized Hooke's law. The shape of the yield surface is given by the Huber–Mises yield criterion, and its evolution, during plastic deformation, is described by the Voce isotropic and Lemaître–Chaboche kinematic hardening laws, due to the mixed plastic hardening behavior exhibited by the two materials, under an associated flow rule. The Voce isotropic and Lemaître–Chaboche kinematic hardening laws are given in Equations (1) and (2), respectively:

$$Y(\overline{\varepsilon}^p) \;=\; Y_0 + (Y_{Sat} - Y_0)[1 - \exp(-C_Y \overline{\varepsilon}^p], \tag{1}$$

where $Y_0$, $Y_{Sat}$, and $C_Y$ are material parameters and $\overline{\varepsilon}^p$ is the equivalent plastic strain.

$$\dot{X} \;=\; C_x \left[ X_{Sat} \frac{\sigma' - X}{\overline{\sigma}} - X \right] \dot{\overline{\varepsilon}}^p, \tag{2}$$

where $\dot{X}$ is the back stress rate, $X_{Sat}$ and $C_x$ are material parameters, $\sigma'$ is the deviatoric Cauchy stress tensor, $X$ is the back stress tensor, $\overline{\sigma}$ is the equivalent stress, and $\dot{\overline{\varepsilon}}^p$ is the equivalent plastic strain rate.

The set of elastic–plastic properties identified, by the fitting of stress-strain curves obtained in smooth specimens, for the AA6082-T6 and AA7050-T6 are shown in Table 1. The AA7050-T6 was found to have a pure kinematic hardening behavior; however, a pure isotropic hardening behavior was also considered for this alloy, as shown in Table 1. The objective was to study the effect of hardening model in the context of the paper.

**Table 1.** Set of elastic–plastic properties.

| Material | $E$ (GPa) | $\nu$ (-) | $Y_0$ (MPa) | $Y_{Sat}$ (MPa) | $C_Y$ (-) | $X_{Sat}$ (MPa) | $C_X$ (-) |
|---|---|---|---|---|---|---|---|
| AA6082-T6 [24] | 70 | 0.29 | 238.15 | 487.52 | 0.01 | 83.18 | 244.44 |
| AA7050-T6 kinematic [25] | 71.7 | 0.33 | 420.50 | 420.50 | 0 | 198.35 | 228.91 |
| AA7050-T6 isotropic | 71.7 | 0.33 | 420.50 | 420.50 | 0 | 0 | 0 |

## 3. Numerical Model

The numerical simulations were carried out with the in-house static implicit finite element code DD3IMP. FCG was studied in single-edge notch tension specimens, Figure 2a, in which the notch depth was kept constant and equal to 8 mm. A crack was placed 96 μm ahead of the notch, leading to an initial crack length, $a_0$, equal to 8.096 mm. Different notch radii, $r$, equal to 8, 4, 2, and 1 mm were modeled, as shown in Figure 2c–f, respectively. As Figure 2 indicates, only $\frac{1}{4}$ of the specimen was modeled considering adequate boundary conditions. A small thickness of 0.1 mm was considered in the simulations, in order to reduce the numerical effort. Both plane stress and plane strain states were

studied, applying proper boundary conditions, as indicated in Figure 2g,h, respectively. In the case of plane strain, an additional boundary condition is necessary to eliminate out-of-plane deformation.

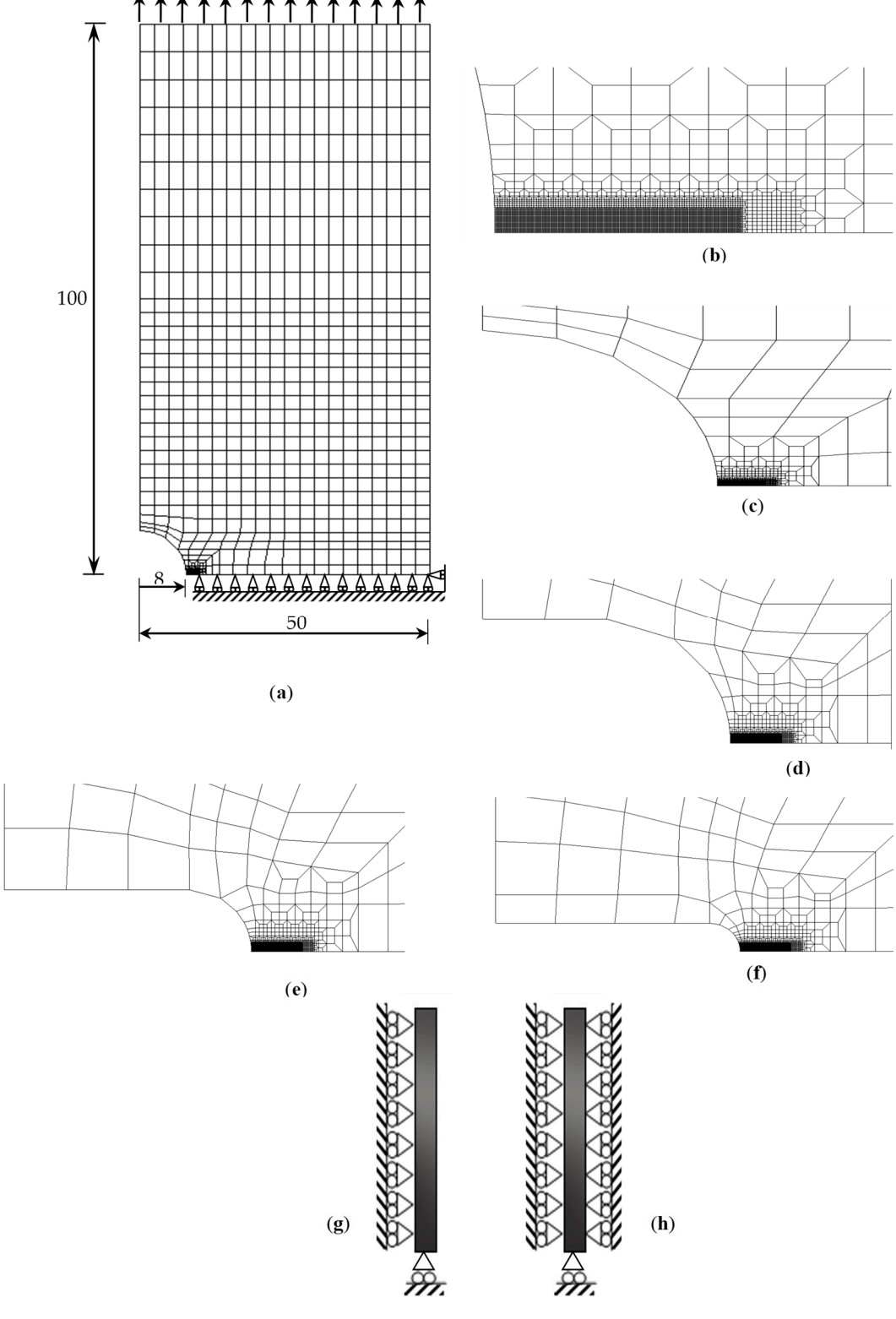

**Figure 2.** Overview of the geometry, finite element mesh, and boundary conditions of the notched specimens. (**a**) Loading, boundary conditions, and main dimensions, in mm. (**b**) Detail of refined mesh. (**c**) $\rho = 8$ mm. (**d**) $\rho = 4$ mm. (**e**) $\rho = 2$ mm. (**f**) $\rho = 1$ mm. (**g**) Plane stress boundary. (**h**) Plane strain boundary conditions.

A cyclic load was applied remotely and normal to the crack growth, as can be seen in Figure 2a, resulting in mode I loading. The maximum and minimum applied loads were $F_{max}$ = 400 N and $F_{min}$ = 4 N, leading to a stress ratio $R$ = 0.01 and to a maximum remote stress of 80 MPa.

A refined mesh was placed at the crack tip region (Figure 2b), with $8 \times 8 \ \mu m^2$ elements, to accurately quantify stress and strain gradients. Out of the crack tip region, a coarser mesh was introduced to reduce the computational cost. The finite element mesh comprised 7175 3D linear isoparametric elements and 7359 nodes. Two load cycles were applied between crack propagations, each one conducting to a crack increment of 8 μm that took place at minimum load. The simulations stopped when the total crack propagation reached 1272 μm, i.e., after 159 crack increments. The contact of the crack flanks was modeled considering a rigid flat surface aligned with the crack symmetry plane. Some simulations were conducted with the possibility to overlap the crack flanks, i.e., for these simulations the flat surface was removed enabling the interference of the crack flanks and disabling crack closure.

To study the effect of the notch on FCG, a comparison was made with an unnotched-cracked specimen. This unnotched model was created by removing the notch in the model presented in Figure 2a, maintaining the crack with $a_0$ equal to 8.096 mm.

The radii considered for the notch in the present work (1–8 mm) are within the values proposed in the literature. Muñiz-Calvente et al. [32] considered radii of 0.4, 1 and 2 mm, while Kolasangiani et al. [33] studied strain ratcheting at notch roots with radius of 1.5, 4.5, and 7.5 mm. Ranganathan et al. [4] considered radius of 3 mm in AA2024-T351 and of 1 mm in AA7749-T7951. Ding et al. [12] considered disk-shaped compact specimens with radii of 0.10, 0.8, and 3.18 mm. The onset of fatigue crack growth is usually defined for a crack length of 0.2–0.25 mm.

## 4. Results

### 4.1. Typical CTOD-F Curves

The present study of FCG is based on the CTOD parameter. Figure 3a presents a classical CTOD versus applied load, F, curve, measured at the first node behind the crack tip. The load cycle starts at minimum load (A) and increases progressively until point B with CTOD remaining constant and equal to zero, i.e., the crack is closed. The crack opens at point B and deforms elastically until point C, which is evidenced by the linear stretch B–C. At point C starts plastic deformation, increasing until reaching the maximum applied load (D). After point D begins the discharge of the specimen. Analogously, first the material exhibits elastic deformation between points D and G. The elastic–plastic transition occurs at point E, and the plastic deformation decreases until reaching (H), where the crack closes. Figure 3b shows plastic CTOD, $CTOD_p$, versus F curve. $CTOD_p$ is obtained from CTOD by subtracting the elastic CTOD, $CTOD_e$. As said before, at stretches B–C and D–G, the material experiences elastic deformation and therefore $CTOD_p$ remains constant. In the elastic–plastic regimes, C–D and G–H, there is a quick increase and decrease, respectively, of the plastic deformation, as shown in Figure 3b. Figure 3a,b show the elastic and plastic CTOD ranges, respectively, $\delta_e$ and $\delta_p$. A linear elastic relation was found between the plastic CTOD range and da/dN [25,28]; therefore, the trends observed of $\delta_p$ are those expected for da/dN.

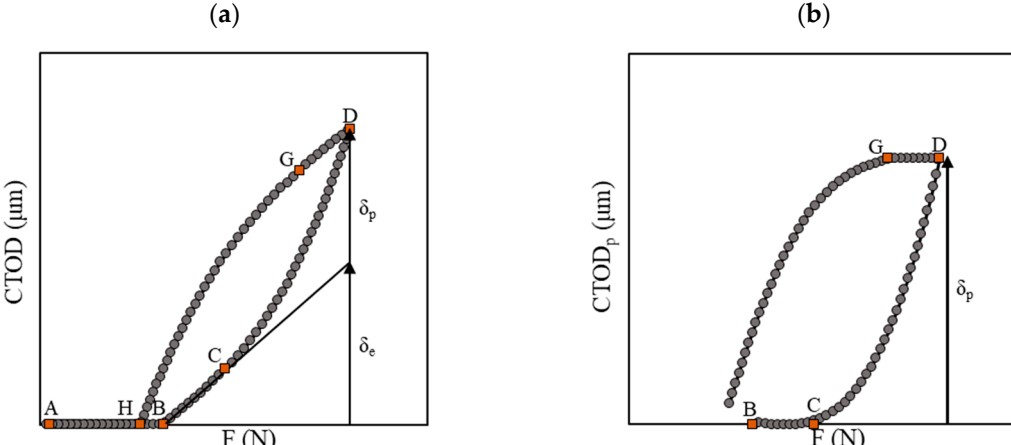

**Figure 3.** Typical plots of crack tip opening displacement (CTOD) versus load, F. (**a**) Total CTOD vs. F; (**b**) Plastic CTOD vs. F (contact).

## 4.2. Unnotched-Cracked Specimen in No Contact Simulations

First, a numerical model containing only the crack was implemented, to make comparisons with numerical data resulting from different notched cracked models. In this first model, the contact of the crack flanks was removed, i.e., the crack closure phenomenon was prevented.

Figure 4a plots $\delta_p$ versus crack propagation, $\Delta a$, for the three material models studied and both plane stress and plane strain conditions. A great influence of material is evident. Higher values of $\delta_p$ are achieved for AA6082-T6, which may be attributed to the lower yield stress of this alloy (see Table 1). Regarding the AA7050-T6, the pure kinematic hardening model provides greater resistance to plastic deformation than the isotropic model, since the level of the stress–strain curve (in uniaxial tension) is higher for the pure kinematic hardening model than for the isotropic hardening model [34]. The effect of stress state varies significantly with material. It has a more pronounced effect in the AA6082-T6 model, where the plane stress case offers less resistance to plastic deformation. For the AA7050-T6 models, there is some influence for small crack increments, which disappears as the crack propagates. The lower values of $\delta_p$ observed for plane strain state compared with plane stress state could be expected considering that stress triaxiality due to plane strain state promotes lower levels of plastic deformation. Concerning the effect of crack increment, there is an initial peak of deformation that results from the material not having previous hardening when first loaded. The application of the following loads produces hardening, which explains the progressive reduction of $\delta_p$. The minimum value indicates the end of this initial transient regime. On the other hand, the increase of crack length increases the stress level ahead of crack tip, which explains the progressive increase of $\delta_p$ with $\Delta a$. The variation is linear with a slope that depends on material, being higher for the AA6082-T6. Figure 4b plots the slope of the linear increase, m, versus $\delta_p$ (measured for a crack increment of 800 µm). The increase of $\delta_p$ promotes a greater influence of crack length, and a linear relation exists between m and $\delta_p$.

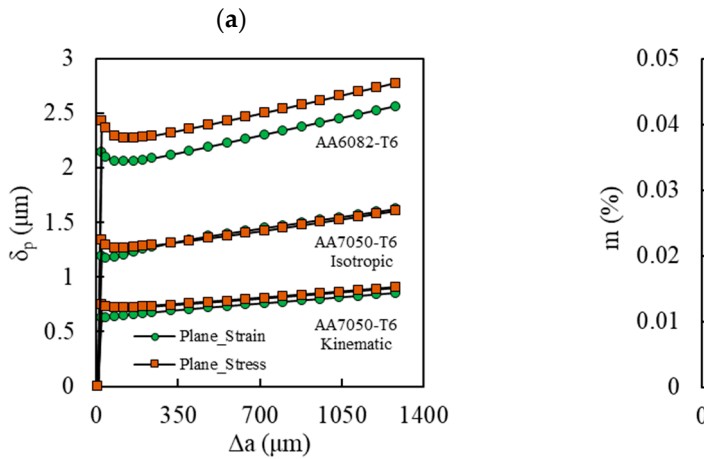
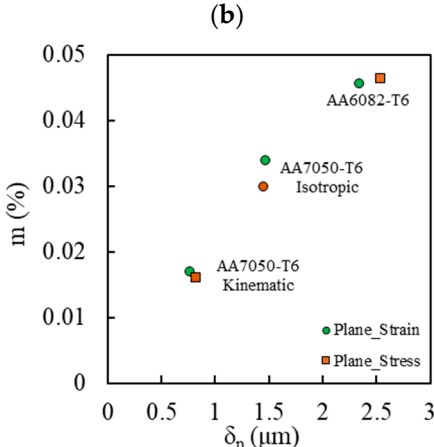

**Figure 4.** Representation of the evolution of (**a**) $\delta_p$ versus $\Delta a$; (**b**) m versus $\delta_p$, with $\delta_p$ evaluated at $\Delta a = 800$ μm (no contact).

### 4.3. Insertion of Contact at the Crack Flanks in the Unnotched-Cracked Specimen

The overlap of the crack flanks was then numerically removed, which enabled the simulation of crack closure. In fact, the phenomenon of crack closure reduces the effective load range, leading to a reduction of elastic and plastic deformations. As previously stated, this phenomenon is expected to have a major effect, particularly in plane stress simulations.

Figure 5a–f compare the evolution of $\delta_p$ against $\Delta a$ with and without contact at the crack flanks for the three materials in study, and plane strain and plane stress conditions. As can be seen, in plane strain conditions (Figure 5a,c,e), the contact simulations have a slight reduction of $\delta_p$, but maintaining the trends observed for the *no contact* simulations. For the plane stress cases (Figure 5b,d,f), the insertion of contact at the crack flanks causes a major reduction of $\delta_p$ with $\Delta a$. The reduction of $\delta_p$ is more pronounced for AA7050-T6 with pure isotropic behavior, and the decrease rate is higher for pure kinematic behavior. On the other hand, the AA6082-T6 shows a relatively slow reduction of $\delta_p$ and the stabilization occurs for higher values of $\Delta a$. The increase of $\delta_p$ with $\Delta a$ observed for the *no contact* simulations is almost eliminated with the contact of crack flanks. Anyway, there is a small increase of $\Delta a$ with $\delta_p$, which is evident in Figure 5d. The minimum values, indicated by small arrows in Figure 5d,f, indicate the end of a transient regime associated with the formation of residual plastic wake. This transient regime is much more extensive than the transient behavior associated with the stabilization of cyclic plastic deformation, observed in Figure 4a.

The variation of crack closure level, quantified by $U^*$, is presented in Figure 6. $U^*$ is given by:

$$U^* = \frac{F_{open} - F_{min}}{F_{max} - F_{min}} \times 100, \tag{3}$$

where $F_{open}$ is the crack opening load and represents the percentage of load cycle during which the crack is closed. The variations of $\delta_p$ observed in Figure 5a,d,f for the simulations with contact, are perfectly symmetric to those observed in Figure 6. This means that the decreases seen in Figure 5 have exactly the same upward trends seen in Figure 6. $U^*$ increases with $\Delta a$ and stabilizes for higher values of $\Delta a$, which explains the reduction of $\delta_p$ with $\Delta a$ and the consequent stabilization. The growth rate of $U^*$ with $\Delta a$ is higher for AA7050-T6 with pure kinematic behavior, and higher values of $U^*$ are achieved for AA7050 with pure isotropic behavior, which are in accordance with Figure 5d,f.

There is a progressive increase of $U^*$ that is linked to the formation of the residual plastic wake, as the crack propagates. The level of crack closure depends on the plastic elongation of the wedges behind crack tip and on crack tip blunting, and these two opposite mechanisms depend on crack tip deformation [35]. The crack propagation required for the stabilization of $U^*$ is greatly dependent on material, being significantly higher for the AA6082-T6. In any case, this stabilization distance is always

significantly higher than that needed to stabilize the cyclic plastic deformation. The stabilized values of $U^*$ are in the range 38–50%, which are typical values for plane stress state. After stabilization there is a small but progressive increase of $U^*$, which explains the lower variation of $\delta_p$ observed in Figure 5d,f, compared to the *no contact* simulations. Under plane strain conditions, the level of crack closure is significantly lower, less than 10%, which explains the small variation observed in Figure 5a,c,e.

　　Figure 6b represents the CTOD versus F curves. The reduction of CTOD and therefore of $\delta_p$ is evident when the contact at the crack flanks is introduced. The effective load range has a major effect on plastic CTOD range.

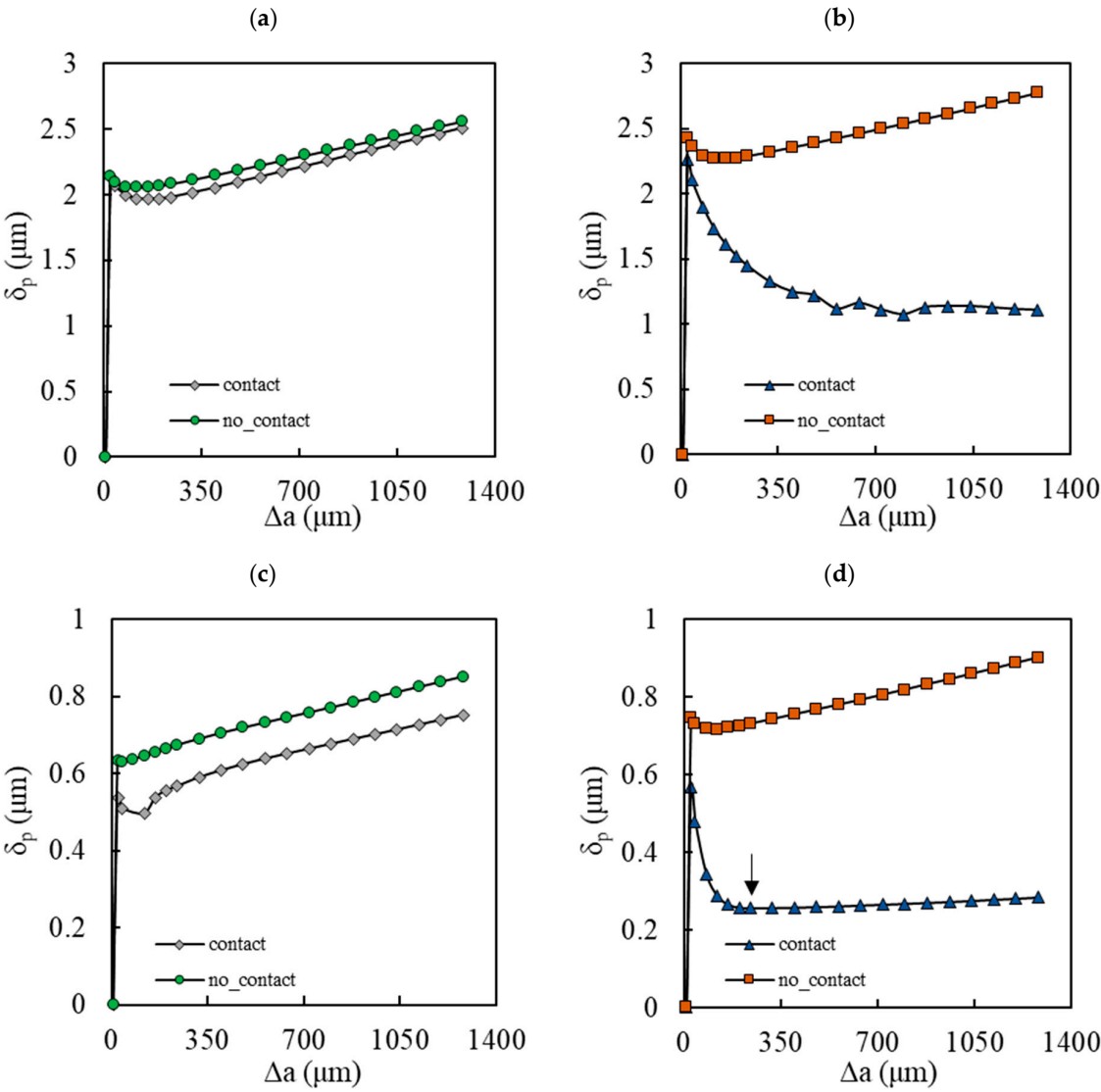

**Figure 5.** *Cont.*

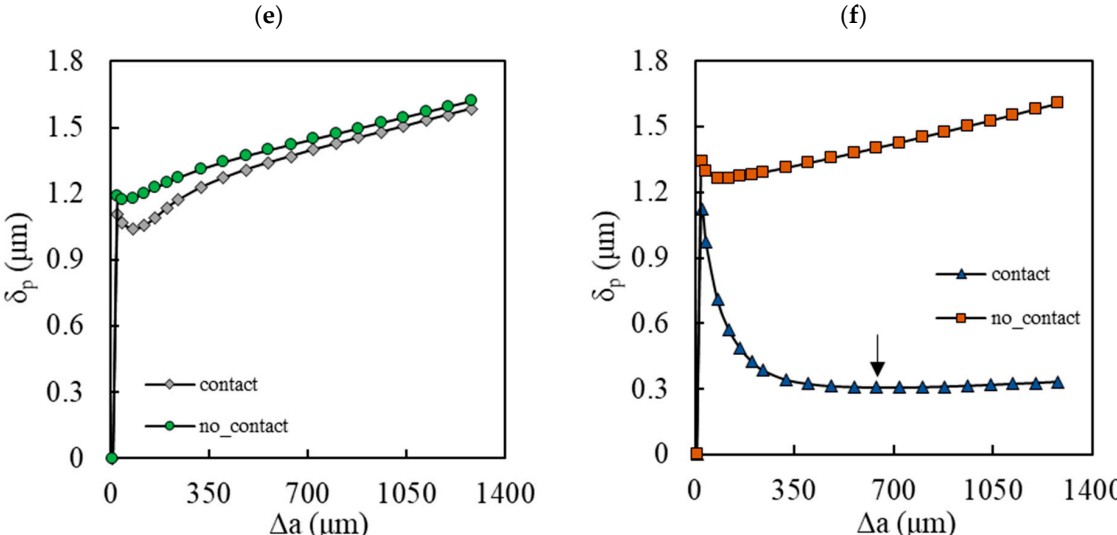

**Figure 5.** Effect of contact of crack flanks on $\delta_p$ versus $\Delta a$ data (unnotched cracked specimen). (**a**) AA6082-T6; plane strain; (**b**) AA6082-T6; plane stress; (**c**) AA7050-T6 with pure kinematic behavior; plane strain; (**d**) AA7050-T6 with pure kinematic behavior; plane stress; (**e**) AA7050-T6 with pure isotropic behavior; plane strain; and (**f**) AA7050-T6 with pure isotropic behavior; plane stress.

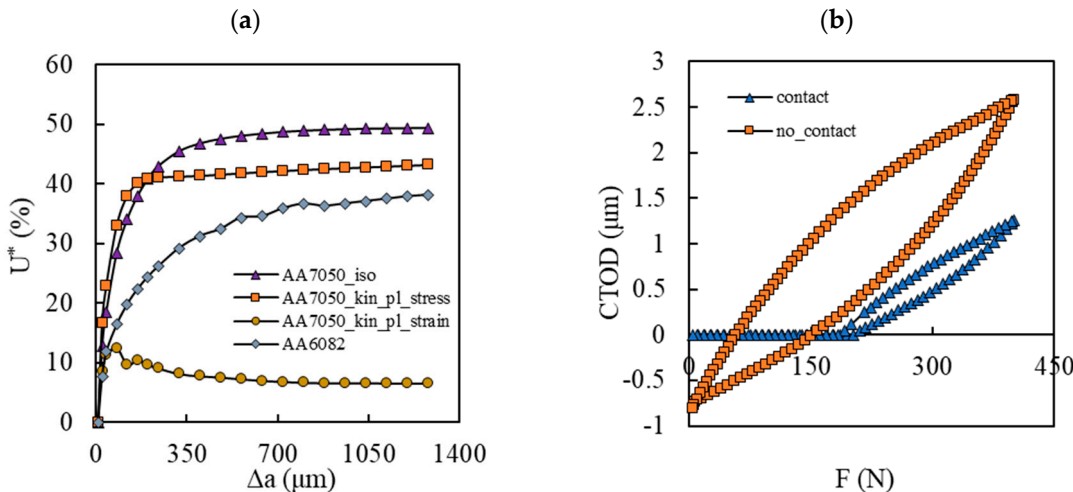

**Figure 6.** (**a**) Evolution of $U^*$ with $\Delta a$, for AA6082-T6 and AA7050-T6, with pure isotropic and kinematic hardening behaviors (unnotched-cracked specimen; contact); (**b**) Effect of $U^*$ on CTOD vs. F curves (AA7050-T6 isotropic; unnotched-cracked specimen; plane stress).

### 4.4. Insertion of a Notch with Different Radius in No Contact Simulations

A notch was placed on the side of the specimen, as illustrated in Figure 2a, with different radii, as schematized in Figure 2b–e. The presence of a notch, as is the case of the crack, is a geometric discontinuity that acts as a stress concentrator. The objective of this section is to study the effect of the conjunction of the notch and crack on FCG, in the *no contact* simulations.

Figure 7a–f compare the development of $\delta_p$ with $\Delta a$ without contact, at the crack flanks, for different notch radii, $r$ = 1, 2, 4, and 8 mm, for the three materials in study assuming both plane strain and plane stress states. The inclusion of the notch reduces the plastic CTOD range, and this effect increases with the radius. This means that the notch has a protective effect on $\delta_p$ and therefore on da/dN, which results from a smoothing effect of stresses around crack tip. There is only one exception for the case of the AA6082-T6 in plane strain state, Figure 7c. The increase in $r$ decreases $\delta_p$, promoting a higher fatigue life as expected since stress-concentration factors decrease with the increase of $r$. The protective

effect of the notch may sound strange. However, note that the crack length of the unnotched crack is 8.096 mm, as illustrated in Figure 8a. The protective effect of the notch is observed comparing the results of cracked geometries exhibited in Figure 8a,b. On the other hand, the comparison between cracked geometries in Figure 8b,c indicates that the notch accelerates da/dN, and this is the comparison typically made in literature. The dashed lines added to Figure 7, indicate the behavior expected for the cracked geometry of Figure 8c, i.e., assuming that the crack length is measured from the notch.

As the crack becomes progressively longer, there is a global trend for the increase of $\delta_p$ with $\Delta a$, being the growth rate higher for lower values of $\Delta a$. The effect of $\Delta a$ is more pronounced for notched specimens than for unnotched specimens. These changes are due to the propagation of the crack tip through the stress field generated by the notch. For relatively large crack increments, the curves start to approach. This means that the notch is losing its effect and also that the difference between the curves is due to the notch. Anyway, for the maximum crack length studied $\Delta a = 1.272$ mm) the effect did not disappear totally because the curves still have some separation. This means that the effect of notch is larger than the 1.272 mm studied, particularly for the higher values of notch radius. On the other hand, for the smallest radius studied ($r = 1$ mm), the curves tend to the results of the unnotched specimens, which means that the boundary of the influence of notch was reached. Without contact at the crack flanks, the trends observed for $\delta_p$ are not affected by stress state or material.

Figure 9 plots $\delta_p$ against $r$ for three values of $\Delta a$, in plane stress state and for the AA7050 with isotropic behavior. The increase of $r$ and $\Delta a$ reduces $\delta_p$, as shown in Figure 7f. The trends observed in Figure 9 suggest an increase of linearity between $\delta_p$ and $r$, as the crack becomes longer. The same relationships were found under plane strain state and for the AA6082 and AA7050 with kinematic behavior. A correlation coefficient squared, $R^2$, of 0.99 was found when a linear trendline was fitted to $\Delta a$ equal to 1272 µm for all cases studied, proving the linearity referred to above.

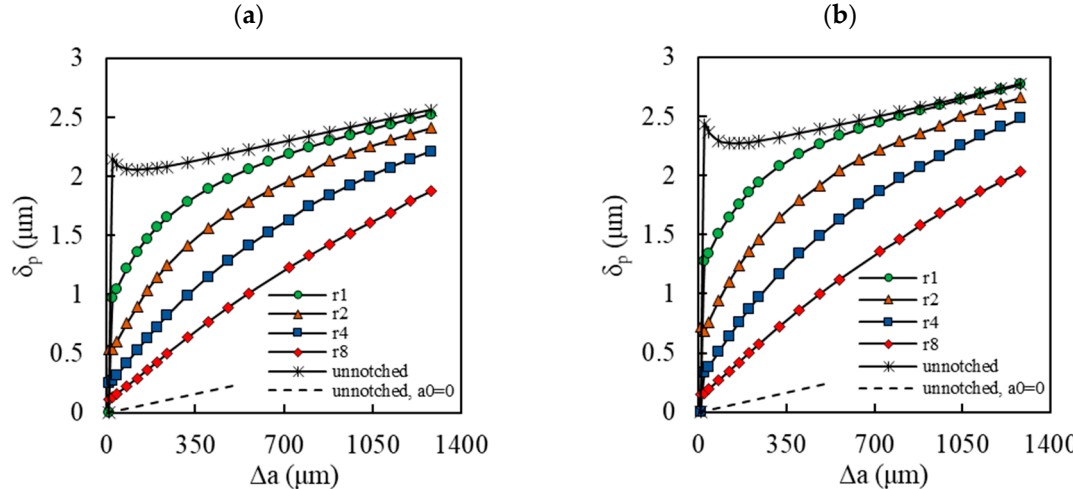

**Figure 7.** *Cont.*

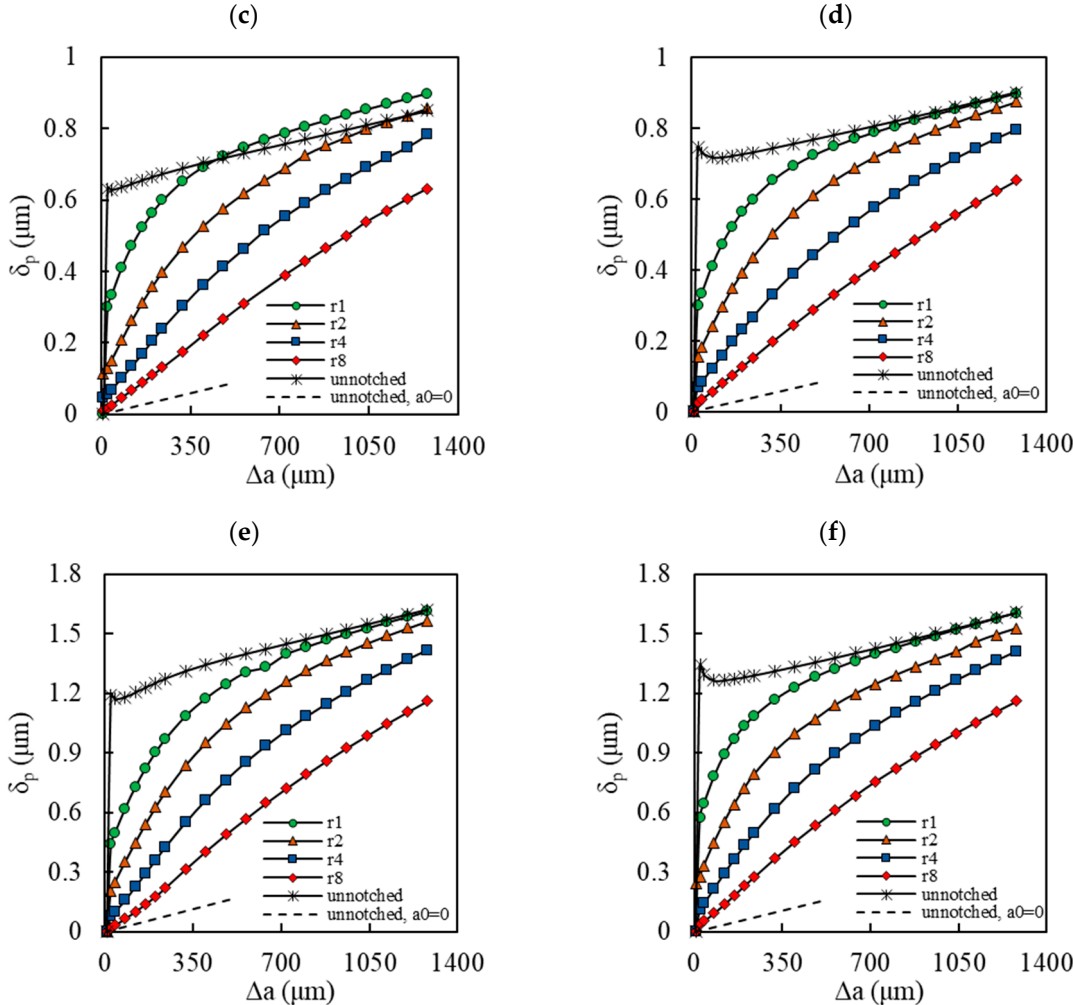

**Figure 7.** Influence of the notch in $\delta_p$ versus $\Delta a$ curves (*no contact*). (**a**) AA6082-T6 in plane strain; (**b**) AA6082-T6 in plane stress; (**c**) AA7050-T6 with pure kinematic behavior in plane strain; (**d**) AA7050-T6 with pure kinematic behavior in plane stress; (**e**) AA7050-T6 with pure isotropic behavior in plane strain; and (**f**) AA7050-T6 with pure isotropic behavior in plane stress.

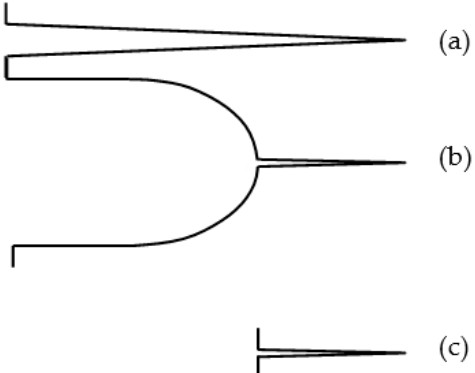

**Figure 8.** Cracked geometries: (**a**) unnotched crack with $a_0 = 8.096$ mm; (**b**) crack with notch; and (**c**) unnotched crack with $a_0 = 0.096$ mm.

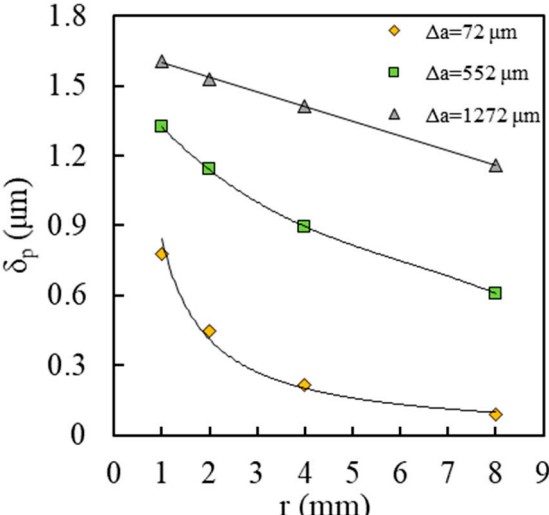

**Figure 9.** Evolution of $\delta_p$ versus *r*, for different crack lengths (AA7050 isotropic; notched-cracked specimen; plane stress).

### 4.5. Insertion of Contact at the Crack Flanks in the Notched-Cracked Specimen

Figure 10a–f show $\delta_p$ against $\Delta a$ for *r* = 1 mm and *r* = 8 mm in contact and no contact simulations. As observed for the unnotched specimens, there is a great influence of crack closure for the plane stress state and a lower effect for the plane strain state. The contact at the crack flanks reduces $\delta_p$, as expected, because there is a reduction of the effective load range. The decrease of notch radius produces a significant increase of crack closure phenomenon in both plane strain and plane stress conditions, being this effect more prominent in plane stress state. This dependence of crack closure on notch radius reduces, apparently, the zone affected by the notches, promoting a faster convergence of the curves for different radius.

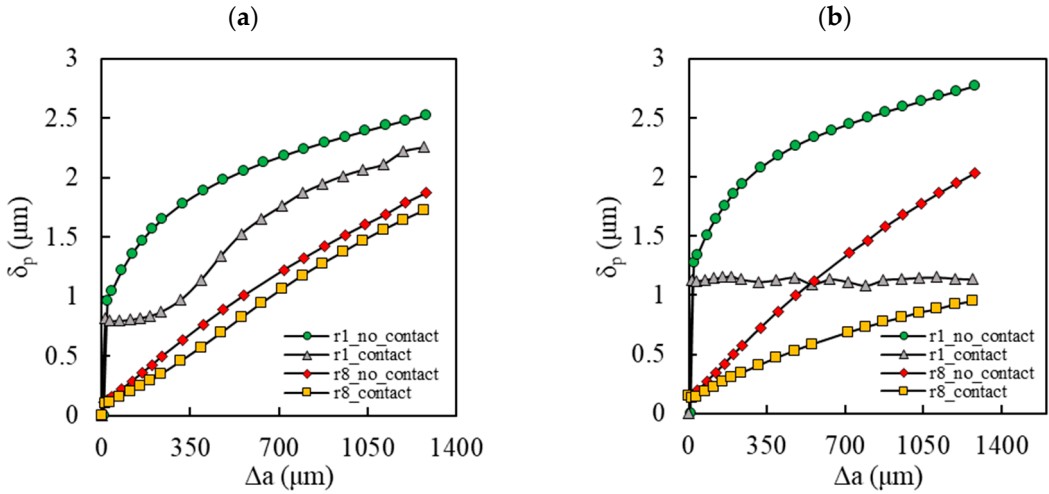

**Figure 10.** *Cont.*

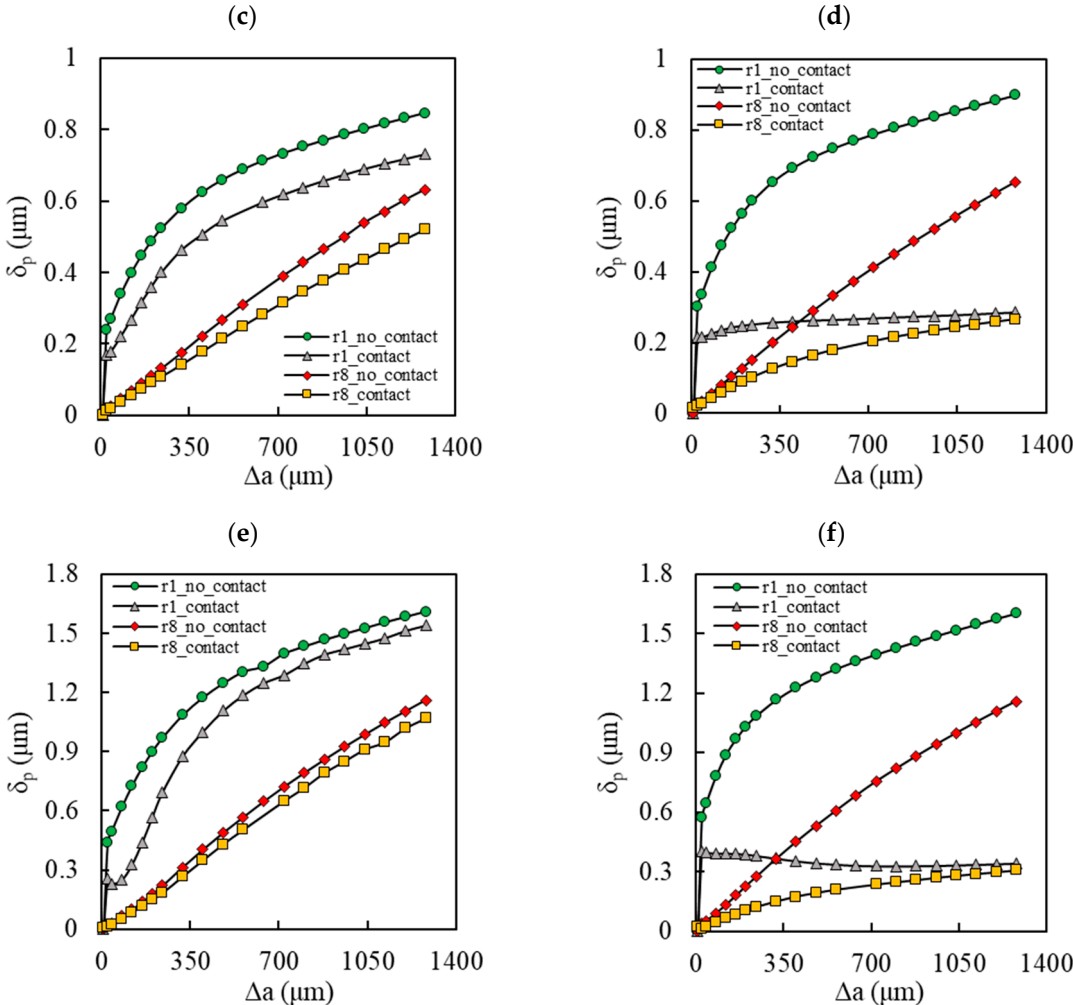

**Figure 10.** Influence of the presence of the notch and the contact at the crack flanks in the $\delta_p$ versus $\Delta a$ curves (notched cracked specimen): (**a**) AA6082-T6 under plane strain; (**b**) AA6082-T6 under plane stress; (**c**) AA7050-T6 with pure kinematic behavior under plane strain; (**d**) AA7050-T6 with pure kinematic behavior under plane stress; (**e**) AA7050-T6 with pure isotropic behavior under plane strain; and (**f**) AA7050-T6 with pure isotropic behavior under plane stress.

*4.6. Comparison of Unnotched and Notched Specimens in Simulations with Contact*

Figure 11a–f plot $\delta_p$ versus $\Delta a$ for simulations with and without notch for the three material models studied here assuming both plane strain and plane stress states. Most of the trends observed without contact (Figure 7) also exist with contact. The notch reduces $\delta_p$, i.e., has a protective effect on da/dN. Once again, the length of unnotched crack used for comparison included the size of the notch. The protective effect increases with notch radius but tends to disappear as the crack propagates from the notch. There is a convergence to the curve corresponding to the unnotched sample, which occurs when the notch effect disappears. However, with contact, there is a marked influence of stress state. For plane strain conditions, there is a sharp increase of $\delta_p$ with $\Delta a$. A relatively large propagation is required to converge the notched and unnotched curves, particularly for the larger values of notch radius (Figure 10a,c,e). Under plane stress conditions, the increase of notch radius reduces substantially $\delta_p$, which means that it has a protective effect. The convergence of the curves is now much faster than for plane strain state, i.e., the notch affected zone seems to reduce substantially. The material does not seem to affect these trends. The V-shaped behavior, typically observed for short cracks growing from

notches, is not observed here, with the exception of AA7050-T6 with pure isotropic behavior and plane stress behavior with notch radius $r$ = 1mm. The increase of load level is likely to promote this behavior.

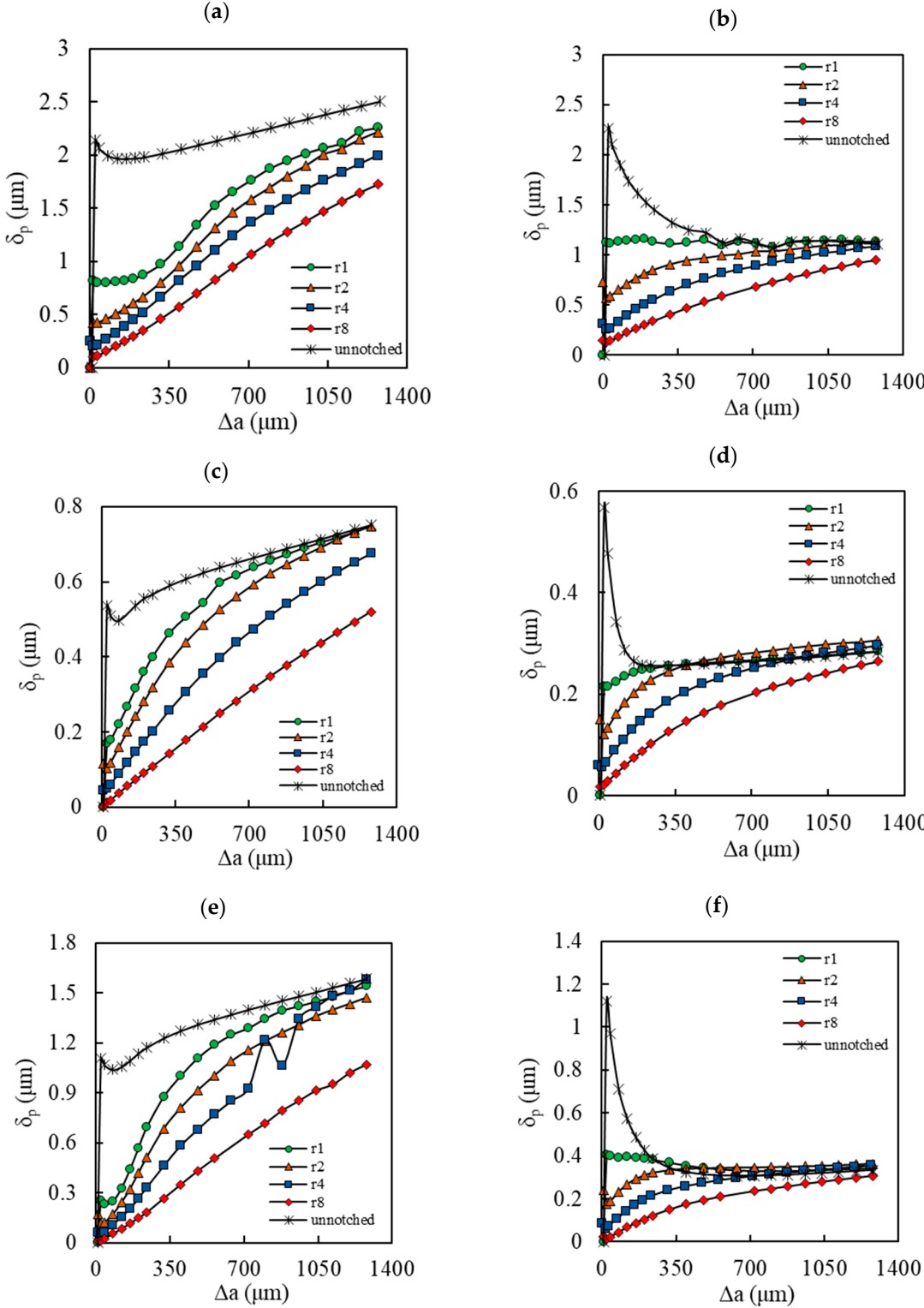

**Figure 11.** Comparison of notched and unnotched geometries regarding the evolution of $\delta_p$ versus $\Delta a$ (notched cracked specimen): (**a**) AA6082-T6 under plane strain; (**b**) AA6082-T6 under plane stress; (**c**) AA7050-T6 with pure kinematic behavior under plane strain; (**d**) AA7050-T6 with pure kinematic behavior under plane stress; (**e**) AA7050-T6 with pure isotropic behavior under plane strain; and (**f**) AA7050-T6 with pure isotropic behavior under plane stress.

$U^*$ was evaluated under plane stress conditions for the three tested material models and for the four notch radii. The development of $U^*$ with $\Delta a$, as exhibited in Figure 12a–c, is similar for all materials. $U^*$ increases as the crack becomes longer, being the rate of increase higher for smaller values of $\Delta a$. As the stress concentrations decrease, i.e., the notch radius increases, lower values of $U^*$ are achieved. The highest values of $U^*$ are observed in Figure 12c for AA7050-T6 with pure isotropic behavior while the lowest are found in Figure 12a for AA6082-T6. Higher values of $U^*$ are reached in the unnotched specimen, except for AA6082-T6. However, the differences relative to the notched cracked specimens tend to be attenuated for higher $\Delta a$. For plane strain state, $U^*$ has a peak for lower values of $\Delta a$ and quickly decreases after. Note that the percentage variation of $\delta_p$ is significantly higher than the percentage variation of $U^*$, because $\delta_p$ increases nonlinearly with effective load range.

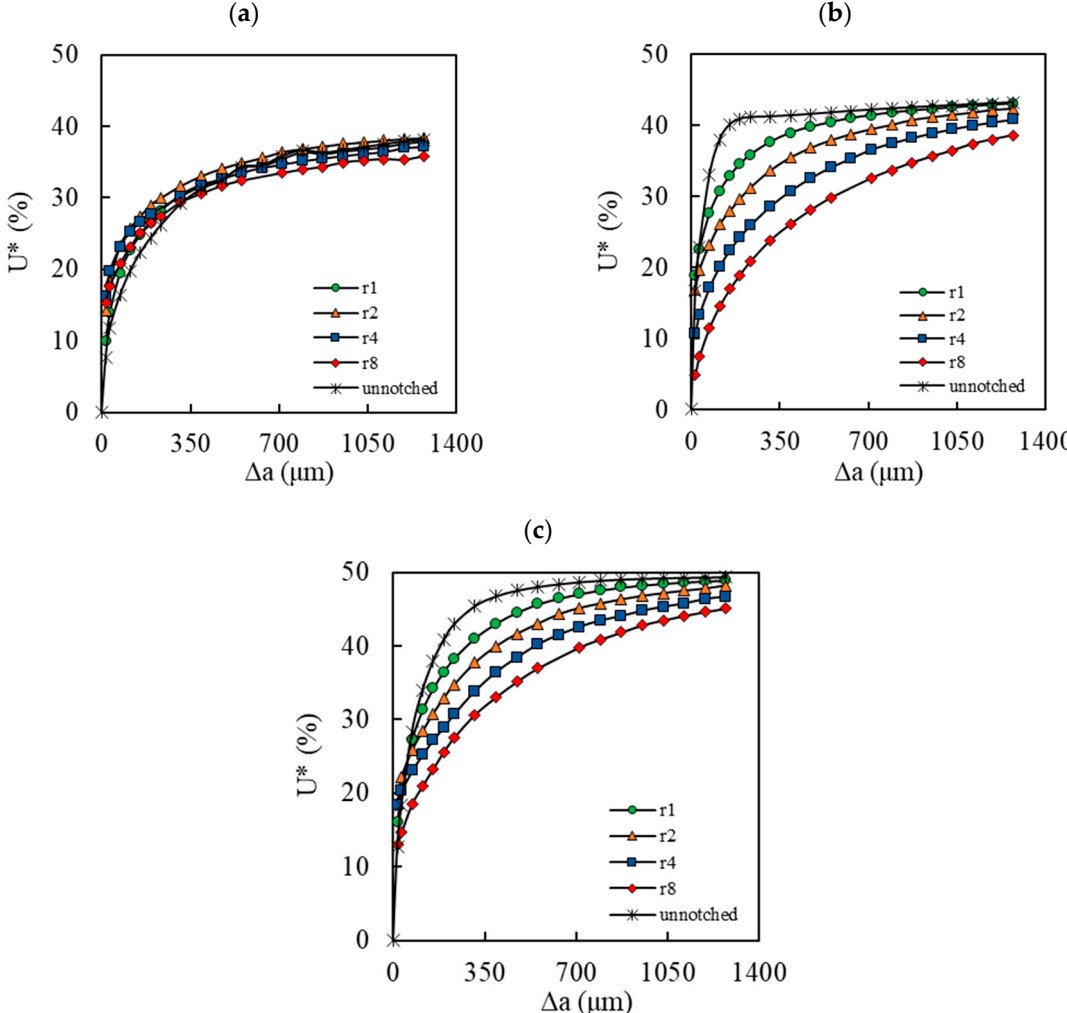

**Figure 12.** Evolution of $U^*$ with $\Delta a$ for different notch radii: (**a**) AA6082-T6; (**b**) AA7050-T6 with kinematic hardening; and (**c**) AA7050-T6 with isotropic hardening (notched cracked specimen, contact; plane stress).

## 5. Discussion

The FCG rate of cracks growing from notches can be explained by the following mechanisms:

- Two transient effects, one associated with the stabilization of cyclic plastic deformation and the other associated with the built of plasticity induced crack closure. The former, which is very evident in unnotched cracked specimens without contact, needs a very short crack propagation to

disappear. The stabilization of crack closure level is much longer, depending on material and increasing with the decrease of notch radius;

- The influence of the notch fields. The reduction of notch radius increases $\delta_p$, and therefore FCG rate, and substantially reduces the extension affected by the notch. Without contact, the limit of notch influence zone has only been reached for the smallest radius studied ($r = 1$ mm);

- The crack closure phenomenon, which has a dramatic effect under plane stress conditions and a limited effect under plane strain conditions. The formation of residual plastic wake is responsible for a progressive increase of $U^*$ towards a stabilized value, after which a slow but steady increase is observed. The long stabilization of the closure values means that in many cases it occupies a significant part of the notch affected zone. The level of crack closure is defined by two opposite mechanisms, which are the elongation of plastic wedges behind crack tip and crack tip blunting. This complexity explains the great influence of material on crack closure level and stabilization distance. The introduction of contact in notched samples has a major effect on plastic CTOD. The decrease of notch radius produces a significant increase of crack closure phenomenon, which apparently attenuates the effect of notch radius and reduces the zone affected by the notch;

- After the notch affected zone there is a linear increase of $\delta_p$ with $\Delta a$, linked to the increase of crack tip stress fields. This effect is clearly observed in unnotched cracked specimens without contact of crack flanks. The slope of the linear variation increases linearly with $\delta_p$.

Figure 13 shows the size of plastic zone size ahead of the notch in uncracked specimens. The plastic zones are larger for plane stress conditions than for plane strain, but are always smaller than 1 mm. The plastic deformation zones extend ahead of initial crack size; therefore, the cracks propagate in plastically deformed material. However, the plastic deformation level is small having a maximum value of 0.93%. Note that the relation between the maximum remote stress and the yield stress is 33.6% for the AA6082-T6 and 11.9% for the AA7050-T6. Naturally, the increase of load level would increase the size of plastic zones and the level of plastic deformation.

| (**a**) | (**b**) |
|---|---|
| 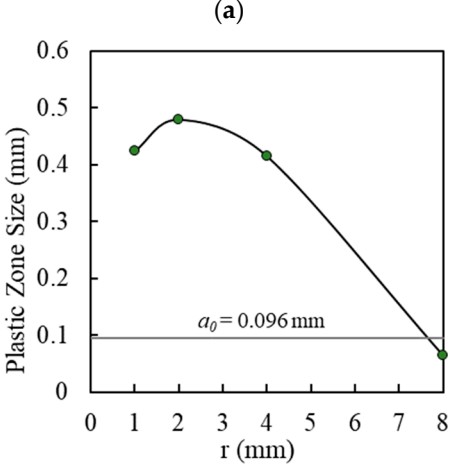 | 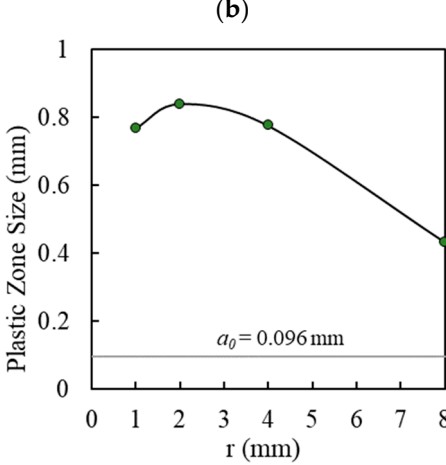 |

**Figure 13.** Plastic zone size versus notch radius, considering $\varepsilon^p = 0\%$ as criterium for the end of the plastic zone (AA6082-T6; contact). (**a**) Plane strain and (**b**) plane stress.

## 6. Conclusions

A numerical approach based on plastic CTOD range, $\delta_p$, was followed to study fatigue crack growth (FCG) from notches. The identification of fundamental mechanisms was made considering notched and unnotched models, with and without contact of crack flanks. Different parameters were studied, namely, notch radius, crack length, and stress state. The main conclusions are:

- in unnotched cracked specimens, there is an initial transient regime associated with the stabilization of cyclic plastic deformation, which slows down the FCG rate to a minimum. After this minimum,

which defines the end of initial regime, the crack growth increases the strength of crack tip fields, producing a linear increase of $\delta_p$. The slope increases with the reduction of yield stress, and a linear trend was observed between the slope and the level of $\delta_p$. The linear increase of da/dN with $\Delta a$ is expected to occur in notched samples after the notch affected zone;

- the introduction of contact produced a major effect on $\delta_p$ in unnotched cracks under plane stress conditions. There is a progressive decrease of $\delta_p$ to a minimum value associated with the formation of residual plastic wake. The minimum value of $\delta_p$ defines the end of the transient regime, which is much more extensive than the transient regime associated with the stabilization of cyclic plastic deformation. The effect of crack closure is very limited under plane strain conditions;
- the notches were found to increase the plastic deformation level at the crack tip. The reduction of notch radius increases the notch effect and reduces significantly the notch affected zone. The limit of this zone was reached only for $r = 1$ mm in this numerical study, being about 0.8 mm for the load level studied. The stress state and material do not affect these trends. As the crack propagates ahead of the notch, a linear relation is observed between $\delta_p$ and notch radius;
- the introduction of contact in notched samples attenuates the effect of notch radius and reduces the notch affected zone;
- concerning the stress state, lower values of $\delta_p$ were obtained for plane strain state compared with plane stress state. This could be expected since the stress triaxiality tends to reduce plastic deformation. Additionally, the effect of stress state is greatly linked with crack closure variations.
- Concerning the material, the global trends related to the influence of notch were found to be independent of material parameters. However, there is a significant influence on crack closure phenomenon. Additionally, there is a global trend for the increase of $\delta_p$ with the reduction of yield stress.

**Author Contributions:** Conceptualization, F.A.; methodology, F.A.; investigation, M.B. and M.C.; data curation, M.B. and M.C.; writing—original draft preparation, M.B. and M.C.; writing—review and editing, R.B. and P.P. All authors have read and agreed to the published version of the manuscript.

**Funding:** This research was funded by the project no. 028789, financed by the European Regional Development Fund (FEDER), through the Portugal-2020 program (PT2020), under the Regional Operational Program of the Center (CENTRO-01-0145-FEDER-028789) and the Foundation for Science and Technology IP/MCTES through national funds (PIDDAC). This research is also sponsored by FEDER funds through the program COMPETE—Programa Operacional Factores de Competitividade—and by national funds through FCT—Fundação para a Ciência e a Tecnologia–, under the project UIDB/00285/2020.

**Conflicts of Interest:** The authors declare no conflict of interest.

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
