# Peer review of "Fatigue Crack Growth from Notches: A Numerical Analysis"

_applsci, doi:10.3390/app10124174_

Round 1

Reviewer 1 Report

The study is interesting and is worth reporting, but the manuscript needs more work. The break down of the manuscript can be done in a much better way to allow better readability and to establish a good flow, which the current manuscript lacks.

Need to define CTOD in the abstract and introduction, before using the acronym. FCG is defined again on line 116, remove the definition.

Referring to the effect of material is ambiguous. Do you mean the choice of material? What aspect of material properties is causing the effect? You need to explain the choice of materials studied (aluminium alloys 6082-T6 and 7050-T6). Why just study AA's?

The choice of including the cracked geometries at the top of some figures, has been very confusing, and can mislead the reader to try to find a correlation between its size and position with the rest of the figure. Proper annotation in the figure caption can replace the cracked geometry sketches at the top.

Figure 7a-f: the dashed straight line is not included in the legends, and whose purpose remains unknown.

The points in the discussion are well-stated, but very hard to wrap one's head around. Could you consider tabulating the results or using a schematic (like in figure 1) to offer a better visual for drawn conclusions.

The effect of varying the notch radius (r) on plastic zone sizes is done for only four values of r (figure 13). Use more values of r for a better representation of the how the plastic zone size varies. Same can be said about the evolution of δp versus r in figure 9.

 Look out multiple errata, please have your manuscript revised thoroughly. Line 429: there is an

Look out for multiple occurrences of random italic words in figure captions following symbols, like the word versus

Author Response

Dear reviewer,

Best regards

Reviewer 2 Report

lines 7-10 are not corrcete: Affilation 2, Affiliation3.....??

line 99: It is wirtten "von Mises". It should be "Huber-Mises"

Figures and captions under this figures should be written in one page. Please chcange Fig.2, 5, 6, 710, 11

Author Response

Dear reviewer,

Best regards

Reviewer 3 Report

Dear Authors,

The paper contains excellent numerical results. There is a lack of experimental data, but accepting the title (and its scope...numerical) formally all is OK. Based on this, I proposed minor revision. During review please consider the points listed below:

  1. Please add affiliation data 2-5
  2. CTOD abbreviation should be initially mentioned in abstract (CTOD – Crack Tip Opening Displacement)
  3. Please add to the table 1 information about fracture toughness – it would be great to know how far from critical load we are during simulations
  4. Generally the presented approach is nice elaborated, However as a Reviewer I feel a lack of the experimental validation of the proposed results. Please provide (if possible) any discussion with link to experimental data in order to validate several parts of numerical analysis.

Author Response

Dear reviewer,

Best regards

Round 2

Reviewer 1 Report

The effort done by the author(s) to address the reviewers remarks is substantial and adequate for publication.